# *Anp32a* Promotes Neuronal Regeneration after Spinal Cord Injury of Zebrafish Embryos

**DOI:** 10.3390/ijms232415921

**Published:** 2022-12-14

**Authors:** Hung-Chieh Lee, Wei-Lin Lai, Cheng-Yung Lin, Chih-Wei Zeng, Jin-Chuan Sheu, Tze-Bin Chou, Huai-Jen Tsai

**Affiliations:** 1Department of Life Science, Fu Jen Catholic University, New Taipei City 242062, Taiwan; 2Institute of Molecular and Cellular Biology, College of Life Science, National Taiwan University, Taipei 10617, Taiwan; 3Institute of Biomedical Science, Mackay Medical College, New Taipei City 25245, Taiwan; 4Liver Disease Prevention and Treatment Research Foundation, Taipei 100008, Taiwan; 5School of Medicine, Fu-Jen Catholic University, New Taipei City 242062, Taiwan

**Keywords:** zebrafish, spinal cord injury, *Anp32a*, neuronal regeneration, proliferation

## Abstract

After spinal cord injury (SCI) in mammals, neuronal regeneration is limited; in contrast, such regeneration occurs quickly in zebrafish. Member A of the acidic nuclear phosphoprotein 32 (*ANP32a*) family is involved in neuronal development, but its function is controversial, and its involvement in zebrafish SCI remains unknown. To determine the role of zebrafish *ANP32a* in the neuronal regeneration of SCI embryos, we microinjected *ANP32a* mRNA into embryos from zebrafish transgenic line *Tg(mnx1:GFP)* prior to SCI. Compared to control SCI embryos, the results showed that the regeneration of spinal cord and resumption of swimming capability were promoted by the overexpression of *ANP32a* mRNA but reduced by its knockdown. We next combined fluorescence-activated cell sorting with immunochemical staining of anti-GFAP and immunofluorescence staining against anti-PH3 on *Tg(gfap:GFP)* SCI embryos. The results showed that *ANP32a* promoted the proliferation and cell number of radial glial cells at the injury epicenter at 24 h post-injury (hpi). Moreover, when we applied BrdU labeling to SCI embryos derived from crossing the *Tg(gfap:GFP)* and *Tg(mnx1:TagRFP)* lines, we found that both radial glial cells and motor neurons had proliferated, along with their increased cell numbers in *Anp32a*-overexpression SCI-embryos. On this basis, we conclude that *ANP32a* plays a positive role in the regeneration of zebrafish SCI embryos.

## 1. Introduction

Spinal cord injury (SCI) is defined as damage to the spinal cord, resulting in the loss of body movement [1]. An estimated 25.5 SCI cases per million people per year are seen in developing countries [2]. SCI can be etiologically divided into traumatic (TSCI) and nontraumatic (NTSCI) types. TSCI is caused by a sudden external mechanical force [3], while NTSCI results from any cause other than trauma [4]. Since patients with SCIs often require intensive medical care, the cost of treatment is comparatively high. Trauma, coupled with treatment cost, is catastrophic for the patient, the patient’s family and, indeed, all of society [5]. Owing to modern medical management, the survival rate and well-being of SCI patients have been substantially improved, yet the availability of medical treatments is still limited, with patients, in some cases, left with nothing but palliative and supportive care and the prospect of a lifelong disability.

SCI can be classified into primary, secondary and chronic phases. Primary injury is caused by mechanical impact to the spine, including contusion, damage to blood vessels and axonal shearing [6]. Immediately after the primary injury, secondary injury results from multifaceted neuroinflammatory responses, including the activation of resident astrocytes and microglia, subsequent infiltration of blood-borne immune cells and oxidative stress, leading to death of functional cells and damage to the tissue microenvironment. Secondary injury can last up to several weeks and spread the lesion to adjacent, otherwise uninjured tissue [7]. The chronic stage of SCI is characterized by the formation of astroglial and fibrous scar tissue around cystic cavitations [3,8,9,10,11]. Consequently, the glial scar and its associated matrix create a cellular and biochemical zone that limits the spread of neuroinflammation from the lesion site to neighboring healthy tissues. However, the presence of glial scar inhibits axonal regeneration/sprouting and cell differentiation in the subacute and chronic phases [12,13]. Therefore, recovery of neural function after SCI in mammals is limited.

Nevertheless, unlike mammalian organisms, many fishes and amphibians, such as zebrafish, can regenerate after traumatic SCI. Traumatic SCI in zebrafish entails the immediate loss of movement from the lesion site to the caudal part. However, in contrast to mammalian paralysis, in which glial scarring restricts spinal cord regeneration, zebrafish glial cells can form a bridge across the epicenter of the SCI to facilitate neuronal proliferation, regeneration and functional restoration within 6 to 8 weeks [13,14,15,16]. Furthermore, glial scar formation of zebrafish is restricted after SCI, favoring rapid axonal regeneration and improvement of neuronal regeneration.

Recently, the recovery functions of many important cellular and molecular factors involved in axonal regeneration have been identified using zebrafish as an in vivo model. For example, Ma et al. (2014) [17] found that the legumain is essential for functional recovery after SCI in zebrafish. Reimer et al. (2008) [18] showed that new motor neurons originating from proliferating Olig2+ ependymo-radial glial cells mature and eventually form synaptic connections. Mokalled et al. (2016) [19] found that connective tissue growth factor is necessary and sufficient to stimulate glial bridging and natural spinal cord regeneration. Wehner et al. (2017) [20] identified that Wnt signaling controls pro-regenerative Collagen XII and thereby contributes to ECM that is growth-promoting for axons at the injury site. Moreover, the absence of immune rejection during transplantation makes zebrafish a promising model for real-time screening of transplanted cells harboring genes which are vital for successful regeneration after SCI [21]. Thus, the use of zebrafish is becoming increasingly widespread for studying neuronal regeneration after SCI.

Previously, we used transcriptomic analysis to screen key regulatory genes involved in neuronal regeneration after SCI of zebrafish embryos [21]. All 3911 candidate genes were categorized into four groups based on gene ontology. In this study, we focused on candidate genes in the regeneration/development-related genetic group and selected *Runx3*, *Elav11A*, *Crabp1a*, *Bace1*, *Inpp5e* and *Anp32a*. Among these candidates analyzed in preliminary studies, only knockdown of either *Inpp5e* or *Anp32a* could significantly impair swimming ability in SCI embryos. Furthermore, the impairment induced by *Anp32a*-MO was more serious than that induced by *Inpp5e*-MO. Therefore, we chose *Anp32a* for further study of the roles of *ANP32a* during neuronal regeneration in zebrafish embryos after SCI.

The acidic (leucine-rich) nuclear phosphoprotein 32 (ANP32) family is comprised of *ANP32a* (also known as PP32, LANP, HPPCn, I1PP2A, MAPM, or PHAPIa), ANP32B (PAL31, APRIL, or PHAPIb), ANP32C (PP32r1), ANP32D (PP32r2) and ANP32E (Cpd1, LANP-L, or PHAPIII) [22,23,24,25], sharing two conserved domains: N-terminal leucine-rich repeat (LRR) domain and C-terminal low-complexity acidic region [26]. *ANP32a* is found in many vertebrates and is involved in diverse physiological functions. For example, *ANP32a* either inhibits phosphatase 2A (PP2A) activity in the regulation of MAP kinases or inhibits acetyltransferase complex in the nucleus to regulate chromatin remodeling and transcription initiation [27,28]. *ANP32a* can associate with MAPs and regulate microtubule function and microtubule-based vesicular trafficking [29]. It also participates in the granzyme A pathway of cell-mediated cytotoxicity [30], promoting caspase-independent apoptosis as a component of the SET complex [31]. Additionally, *ANP32a* plays important roles in a variety of human pathophysiological processes. For example, it acts as a tumor suppressor in mammalian cells by causing cell cycle arrest and cell growth inhibition [32,33,34]. In contrast, Yan et al. (2017) [35] reported that *ANP32a* promotes colorectal cancer proliferation by inhibition of p38 and activation of Akt signaling pathways. Tian et al. (2021) [36] also found that *ANP32a* increases the proliferation, migration and invasion of human hepatocellular carcinoma cells by activating the HMGA1/STAT3 pathway.

In neuronal cells, CXCL12-mediated regulation of *ANP32a* is implicated in the regulation of neuronal survival [37]. *ANP32a* reduction may contribute to abnormal neuritic morphology in the dominantly inherited neurodegenerative disorder, spinocerebellar ataxia type 1 [38]. In contrast, Chen et al. (2008) [39] reported that increased *ANP32a* impairs microtubule network and neurite outgrowth in the brains of Alzheimer’s patients. Thus, *ANP32a* knockdown could rescue synapse and memory loss via chromatin remodeling in an Alzheimer’s model [40]. Based on the many studies shown above, it can be concluded that *ANP32a* may play different functional roles in different physiological processes and in different cell types. Nonetheless, it remains controversial in terms of favoring cell differentiation or proliferation. Moreover, the involvement of *ANP32a* in zebrafish SCI remains largely unknown.

In this study, employing gain- and loss-of-function strategies on zebrafish transgenic line *Tg(mnx1:GFP)*, in which GFP is specifically labeled in motor neurons, we found that overexpression of *Anp32a* enhanced the regeneration of motor neurons and swimming capability in SCI zebrafish larvae. In contrast, knockdown of *Anp32a* reduced the regeneration of motor neurons and swimming capability. When we applied immunofluorescence staining, combined with fluorescence-activated cell sorting (FACS), we found that *ANP32a* could facilitate cell proliferation after SCI. Furthermore, taking advantage of zebrafish transgenic line *Tg(gfap:GFP*), in which GFP is specifically expressed in radial glial cells, we found that *ANP32a* promotes the proliferation of radial glial cells. Increased radial cells and motor neurons derived from increased motor neuron progenitors aid in the neuronal regeneration of zebrafish after SCI.

## 2. Results

### 2.1. Optimal Recovery Time after SCI as a Baseline to Study Spinal Cord Regeneration of SCI Embryos

Neuronal regeneration occurs rapidly and dynamically, but it can still be observed directly on the transected spinal cord of zebrafish embryos by intravital microscopy [41,42,43]. To investigate the dynamic changes of transected spinal cord during the process of regeneration after SCI, we employed zebrafish embryos from transgenic line *Tg(mnx1:GFP)*, in which GFP is expressed in neuronal cells. The spinal cord, i.e., between the 17th and 20th somite located above the anus of embryos from *Tg(mnx1:GFP)* at 48 hpf, was transected by tweezer. Fluorescent images of embryos at both sides of the SCI adjacent area could be captured after SCI at each subsequent interval (Figure 1A). Compared to untreated embryos, a continuous expression of GFP signal along the spinal cord was interrupted at the SCI site at 0 h post-injury (hpi), indicating that a spinal cord segment at the lesion area had been traumatized by mechanical damage (Figure 1A). However, as shown in Figure 1A, some axons were seen entering the lesion site, and axonal regrowth had occurred by 12 hpi. Large bundles of longitudinally oriented axons were seen bridging the lesion site at 24 hpi, and significant numbers of regenerated axons were observed to cover the epicenter of the SCI site at 36 hpi. These data show that the robust and rapid regeneration of spinal cord in SCI zebrafish embryos could be easily and clearly traced.

When we used the Zen 2010b Sp1 image acquisition and analysis software, we found that a fluctuating intensity of the GFP signal in the area adjacent to the SCI site (caudal and rostral sides) could be detected, but that this dropped off toward the SCI epicenter (Figure 1B). Drop-off points, which were located at both sides, are visualized as yellow spots in Figure 1B, but these could also be measured by sensitive instrumentation, which indicated that the distance was longer than that between the two broken lines observed by the naked eye (yellow spots vs. broken line, Figure 1B). However, as shown in Figure 1C, longer time after SCI correlated with shorter distance between the two drop-off points. More specifically, we found that the slope of decreasing distance sharpened between 24 and 36 hpi (Figure 1C), suggesting that axonal regeneration might become more active at 24 hpi. Thus, in this study, we focused our attention on the effect of *ANP32a* on spinal cord regeneration of SCI *Tg(mnx1:GFP)* embryos at 24 hpi.

### 2.2. Effect of Knockdown or Overexpression of ANP32a mRNA on the Expression of ANP32a Protein in Zebrafish Embryos

To study the effect(s) of *ANP32a* on axonal regeneration of SCI zebrafish embryos, we employed embryos from the *Tg(mnx1:GFP)* transgenic line in which GFP is specifically labeled in motor neurons. For our loss-of-function strategy, we designed an antisense morpholino, *Anp32a*-MO (Figure 2A), to block the translation of endogenous *Anp32a* mRNA, while for our gain-of-function strategy, we designed a wobble-*Anp32a*-flag mRNA which could not be completely base-paired with *Anp32a*-MO (Figure 2A). Consequently, *ANP32a* could be translated when the rescue experiment was performed through simultaneous microinjection of *Anp32a*-MO and wobble-*Anp32a*-flag mRNA. Based on Western blot analysis, the base level of endogenous *ANP32a* was faintly presented in the untreated control embryos but completely absent in the *Anp32a*-MO-injected embryos (Figure 2B). Interestingly, *ANP32a* was significantly increased in embryos injected with wobble-*Anp32a*-flag mRNA only, as well as wobble-*Anp32a*-flag mRNA combined with *Anp32a*-MO (Figure 2B), suggesting that the loss of *ANP32a* caused by *Anp32a*-MO could be rescued by additive injection of wobble-*Anp32a*-flag mRNA.

### 2.3. Using Gain- and Loss-of-Function Strategies to Analyze the Effects of ANP32a on Axonal Regeneration of SCI-Zebrafish Embryos

To study the role of *ANP32a* on the spinal cord regeneration process after SCI, we injected either *Anp32a*-MO or wobble-*Anp32a*-flag mRNA at the one-cell stage, followed by SCI at 48 hpf. We then analyzed axonal regeneration at 18 and 24 hpi (Figure 3A). Using Zen 2010b Sp1 software, the GFP fluorescent signal around the lesion area of SCI, as selected from the ROI information, was recorded. The results showed that the otherwise continuous GFP signal was interrupted at 0 hpi, as marked by two drop-off points, shown as yellow dots. Figure 3B (left column) shows the fluorescence situated outward from the SCI epicenter at 18 hpi but then moving inward and consolidating at 24 hpi.

For the gain-of-function approach, the inward movement of GPF signals from the drop-off points toward the SCI epicenter of SCI embryos injected with wobble-*Anp32a*-flag mRNA was faster than that of the control group consisting of SCI embryos without wobble-*Anp32a*-flag mRNA injection (Figure 3B). Using ImageJ v1.53t software, we quantified the intensity of GFP fluorescence presented within the wider radius of the SCI area in SCI embryos. The results demonstrated that the intensity of the GFP signal at the SCI epicenter in *Anp32a*-overexpressed SCI embryos was higher than that of embryos following SCI only (Figure 3C).

For the loss-of-function approach, the inward movement of GFP signals from the drop-off points toward the SCI epicenter of SCI embryos injected with *Anp32a*-MO was slower than that of SCI embryos (Figure 3D). Using ImageJ software, we also quantified the intensity of GFP fluorescence presented within the wider radius of the SCI area. The results demonstrated that the intensity of the GFP signal at the SCI epicenter in *Anp32a*-knockdown SCI embryos was lower than that of embryos following SCI only (Figure 3E).

Taken together, both gain- and loss-of-function evidence strongly suggest that *ANP32a* plays a positive role in axonal regeneration in spinal cord after SCI in zebrafish embryos.

### 2.4. Using Gain- and Loss-of-Function Strategies to Analyze the Effects of ANP32a on Swimming Capability of SCI Zebrafish Embryos

Next, we performed a functional recovery analysis of larvae developed from embryos treated with gain- or loss-of-function of *ANP32a*, followed by SCI, according to the timetable illustrated in Figure 4A.

For the gain-of-function approach, the swimming distances of zebrafish larvae derived from untreated control, SCI embryos and *ANP32a*-overexpressed SCI embryos were compared. We observed that the swimming behavior responded to head touch [44], and we recorded the swimming route of SCI embryos at 24 hpi, i.e., at 72 hpf (Figure 4B). After calculating swimming distance, we found that the swimming capability of larvae developed from SCI embryos, compared with uninjured larvae, was impaired (Figure 4C). However, the locomotive function of SCI larvae injected with wobble-*Anp32a*-flag mRNA at the one-cell stage was significantly improved (Figure 4C).

Similar to the procedures described above, we performed the loss-of-function approach of *ANP32a* to determine its role in spinal cord regeneration. Compared with SCI larvae, the swimming capability of larvae developed from SCI and *Amp32a*-MO-injected embryos was less competent (Figure 4B,C), but could, nonetheless, be completely rescued by injection with wobble-*Anp32a*-flag mRNA (Figure 4B,C).

Based on the evidence obtained from the swimming behavior after the gain- and loss-of-function approaches, as shown above, we concluded that *ANP32a* improves neuronal regeneration in larvae after SCI, playing a positive role in spinal cord regeneration of SCI zebrafish larvae.

### 2.5. ANP32a Played a Positive Role in the Proliferation of Neural Cells at the SCI Site during Regeneration of SCI Embryos

Thus far, it has been established that the overexpression of *ANP32a* improves neuronal regeneration after SCI in zebrafish embryos, strongly indicating that *ANP32a* plays a positive role in spinal cord regeneration. Therefore, we hypothesized that *ANP32a* may confer neural cells with increased proliferation during the regeneration of SCI embryos. To prove this hypothesis, we employed immunofluorescence staining on embryos using antibody against proliferation marker phosphor-histone 3 (PH3). Embryos at the one-cell stage were injected with *Anp32a* mRNA, following SCI at 48 hpf, and immunostained with anti-PH3 at the adjunct of the SCI site at 18 and 24 hpi, according to the timetable shown in Figure 5A. Compared to SCI embryos, which served as the control group, the number of PH3-positive cells presented at the SCI site of *Anp32a* mRNA-injected embryos was increased by 2.32- and 1.53-fold at 18 and 24 hpi, respectively (Figure 5B,C). This line of evidence suggested that the proliferation of neural cells at the SCI site had increased in SCI embryos overexpressing *Anp32a* during regeneration.

### 2.6. ANP32a Improved the Proliferation of Motor Neuron Progenitor Cells and Radial Glial Cells at the SCI Site of SCI Embryos

The above results demonstrate that *ANP32a* increases the proliferation of neural cells during the regeneration of SCI embryos. We next investigated which cell types were induced to the proliferative state by *ANP32a* during regeneration after SCI, since many cell types are involved in neuronal repair after SCI in zebrafish [45]. To address this question, we performed further immunohistochemical staining on SCI-*ANP32a*-overexpressing *Tg(gfap:GFP)* embryos using antibody against Olig2, a cell marker that identifies motor neuron progenitor cells which are able to differentiate into motor neurons [46], and 4′,6-diamidino-2-phenylindole (DAPI) staining to quantify DNA for cell cycle analysis. After staining, we carried out FACS and collected Olig2-postive cells to analyze their cell cycle phase. Based on scatter profiles obtained from FACS (Figure 6A), the percentage of motor neuron progenitor cells among total cell population in the SCI embryo control group at 24 hpi was 9.6% (P2 value shown in Figure 6B), while the P2 value of SCI-*ANP32a*-overexpressing embryos was 11.1% (Figure 6B), indicating that the number of motor neuron progenitor cells had increased, along with the overexpression of *ANP32a*. In a parallel experiment, we performed cell cycle analysis, as shown in Figure 6C. The results showed that the percentage of S + G2 phase of SCI embryos (control group) was 40.17% (15.13% in S phase and 25.04% in G2 phase), while that of SCI-*ANP32a*-overexpresssing embryos was 23.34% (11.69% in S phase and 11.65% in G2 phase) (Figure 6C). These results mean that the proliferation of motor neuron progenitor cells occurred prior to 24 hpi. Therefore, while the percentage of motor neuron progenitor cells did increase, the percentage of S + G2 phase cells did not increase in *ANP32a*-overexpressing SCI embryos at 24 hpi. This could mean that motor neuron progenitor cells were induced to increase by *ANP32a* by 24 hpi in the SCI-*ANP32a*-overexpressing embryos, but that many proliferated motor neuron progenitor cells in the SCI-*ANP32a*-overexpressing embryos had already stopped undergoing proliferation at 24 hpi. If the latter supposition is true, it means that these proliferated motor neuron progenitor cells induced by *ANP32a* might start to differentiate into motor neurons at 24 hpi.

Using the same strategy as that described above, we employed zebrafish transgenic line *Tg(gfap:GFP)*, in which radial glial cells were specifically labeled with GFP. Based on the scatter profiles obtained from FACS analysis (Figure 7A), the percentage of radial glial cells relative to the total cell population of the SCI embryo control group at 24 hpi was 13.8% (P2 value shown in Figure 7B), while the P2 value of SCI-*ANP32a*-overexpressing embryos was 15.1% (Figure 7B), indicating that the number of radial glial cells had increased in the SCI-*ANP32a*-overexpressing embryos. Meanwhile, based on the cell cycle analysis, as shown in Figure 7C, we found that the percentage of S + G2 phase presenting in the radial glial cells of SCI embryos (control group) at 24 hpi was 6.83% (3.94% in S phase and 2.89% in G2 phase), while that of SCI-*ANP32a*-overexpressing embryos was 8.24% (5.21% in S phase and 3.03% in G2 phase), indicating that the proliferation of radial glial cells was actively progressing in the *ANP32a*-overexpressing SCI embryos at 24 hpi. Taken together, we hypothesized that *ANP32a* contributes to the proliferation and increase of radial glial cells at 24 hpi.

To further test the above hypothesis, we used immunofluorescent staining by antibodies against PH3 in SCI-*Tg(gfap:GFP)* embryos at 24 hpi (Figure 8A). After quantifying the number of GFAP-positive cells (green) colocalized with PH3-positive cells (red), we demonstrated that the number of mitotic radial glial cells was significantly increased in the *ANP32a*-overexpressing embryos at the SCI lesion site, compared to that of the SCI control embryos at 24 hpi (Figure 8B), suggesting that the radial glial cells induced by *ANP32a* were undergoing active proliferation. Next, we injected BrdU into the brain and then performed SCI at 48 hpf for embryos derived from the cross strain between *Tg(gfap:GFP)*, in which radial glia cells were tagged by GFP, and *Tg(mnx1:TagRFP)*, in which motor neurons were tagged by RFP (Figure 8C–F). We found that the number of cells displaying a BrdU-positive signal (purple) and colocalized with GFAP-positive (green, Figure 8C,D) and mnx-positive (red, Figure 8E,F) signals was increased in SCI embryos overexpressing *ANP32a* at the SCI site. These results suggest that *ANP32a* overexpression increases the numbers of both radial glial cells and motor neurons, which may contribute to spinal cord regeneration of SCI embryos.

## 3. Discussion

Using a zebrafish model of SCI in this work, we reported that zebrafish *ANP32a* promotes the proliferation of radial glial cells and motor neurons at the epicenter of the SCI site in embryos at 24 hpi, regenerating neuronal function and rescuing swimming capacity. Therefore, we propose that zebrafish *ANP32a* plays a positive role in the regeneration of zebrafish embryos after SCI.

### 3.1. ANP32a Promotes Cell Proliferation in Some Cellular Processes, including Regeneration of SCI Zebrafish Embryos

*ANP32a* has been proposed as an inhibitor of tumor growth in pancreatic cancer, prostate cancer and lung cancer [47,48,49,50]. On the other hand, some studies have reported that *ANP32a* promotes the development of colorectal cancer, glioma and leukemia [51,52,53,54]. Tian et al. (2020) [36] further found that the expression of *ANP32a* is increased in hepatocellular carcinoma patients, since *ANP32a* promotes the proliferation, migration and invasion of hepatocellular carcinoma cells through the HMGA1/STAT3 pathway. *ANP32a* expression is also upregulated in glioma patients, promoting the proliferation of glioma cells through the Akt/p27/stathmin pathway [52]. Earlier, however, Kadota and Nagota (2011) [28] demonstrated that *ANP32a* is required for positive transcriptional control of type I IFN-stimulated genes by its ability to promote cell proliferation. Thus, it appears that the role of the nuclear protein *ANP32a* in different cellular processes remains controversial.

Here, we applied both FACS and IHC techniques to zebrafish embryos to explore the effects of *ANP32a* on the neuronal processes involved in spinal cord regeneration after SCI. In this study, we provided specific evidence to demonstrate that overexpression of *ANP32a* in SCI embryos results in: (1) the proliferation of radial glial cells and motor neuron progenitors, which occurred at 24 hpi, while the proliferation of motor neuron progenitor cells occurred prior to 24 hpi; and (2) a significant increase in the total numbers of motor neurons and radial glial cells at 24 hpi. Therefore, we can safely conclude that *ANP32a* promotes the proliferation of neural cells in the spinal cord during regeneration after SCI in zebrafish. In hepatocellular carcinoma and glioma, it was discovered that *ANP32a* promotes cellular proliferation [36,52]. Similarly, we found that *ANP32a* promotes the proliferation of these neural cells in SCI embryos at the adjunct lesion site of the spinal cord to hasten recovery after SCI.

Spinal cord injury leads to the influx of a significant number of non-neural cells to the injury site. The results shown in Figure 5 may indicate that neural cells proliferate after SCI. To confirm this hypothesis, we performed FACS. The results shown in Figure 6 and Figure 7 demonstrated that *ANP32a* could improve the proliferation of motor neuron progenitor cells and radial glial cells at the SCI site of SCI embryos, confirming our speculation. Furthermore, as shown in Figure 8, the overexpression of *ANP32a* in SCI embryos from crossing *Tg(gfap:GFP)*, in which radial glial cells were tagged by GFP, and *Tg(mnx1:TagRFP)*, in which motor neurons were tagged by RFP, displayed an increase of both GFP +/BrdU+ and RFP +/BrdU+ signals at the lesion site, compared to control embryos. This evidence suggested that, at the very least, motor neurons and radial glial cells were actively proliferated at the injury site and that such an increase was induced by *ANP32a* overexpression. The proliferation of non-neural cells owing to *ANP32a* overexpression may be an avenue of investigation in the future.

Olig2 labels not only motor neuron progenitors but also oligocytes and oligodendrocyte precursor cells. However, to further confirm our hypothesis that *Anp32a* overexpression induces an increase of motor neurons, we injected BrdU into the brain and then performed SCI at 48 hpf for embryos derived from the cross strains described above. As shown in Figure 8, the number of cells displaying a BrdU-positive signal (purple) and colocalized with GFAP-positive (green, Figure 8C,D) and mnx-positive (red, Figure 8E,F) signals was increased in SCI embryos overexpressing *ANP32a* at the SCI site, suggesting that *ANP32a* overexpression increased the numbers of both radial glial cells and motor neurons. Therefore, we conclude that the overexpression of *ANP32a* promotes the proliferation of motor neuron progenitors and radial glial cells, which, in turn, increases the number of motor neurons. This outcome may contribute to neuronal regeneration in the spinal cord of SCI embryos.

Specifically, we noticed that radial glial cells had actively proliferated and significantly increased in number at the injury epicenter of SCI embryos at 24 hpi. It has been reported that radial glial cells are highly responsive to SCI stress and are the major cellular constituent able to initiate the formation of a glial bridge between the caudal and rostral sides of the SCI site, mediating spinal cord regeneration in zebrafish at 24 hpi [21,55]. Therefore, the active proliferation and significant increase of radial glial cells might play an important role in the regeneration of SCI embryos at the initial (i.e., 24 hpi) stage. Although only two cell types were presented in this study, it is reasonable to speculate that other cell types, such as neural stem progenitor cells, might also proliferate and increase to, finally, play their own roles in the regeneration of SCI embryos.

### 3.2. The Multifunctional Protein ANP32a Is Also Involved in Neurogenesis, including Neuronal Development and Neural Regeneration of Spinal Cord after SCI

Based on previous literature, we have seen that *ANP32a* is a multifunctional protein involved in the regulation of many cellular processes, including tumor suppression, apoptosis, cell cycle progression, neuritogenesis and transcriptional regulation [56]. Indeed, it has been suggested that *ANP32a* plays an important role in mammalian nervous system development and differentiation. For example, compared to the fetal brain, Wang et al. (2015) [56] reported that *ANP32a* is more highly expressed in the adult brain of humans and mice. *ANP32a* is abundantly expressed in cerebral cortex and the granular layer and Purkinje cells in cerebellum. In both cases, increased expression level is dependent on developmental stages. This evidence suggests that *ANP32a* plays a positive role in the brain development of mammals, based on the differentiation of neurons in the cerebellum and cerebral cortex. Furthermore, Opal et al. (2003) [29] found that *ANP32a* is located in the nucleus of undifferentiated Neuro2a cells but tends to be drawn to the cytoplasm to interact with the light chain of microtubule-associated protein 1B in order to impact neurite outgrowth during neuritogenesis. This evidence suggests that the distribution and subcellular localization of *ANP32a* may affect different cellular processes through interaction with microtubule-related proteins in neural development.

*ANP32a* is also a positive factor in the Spinocerebellar ataxia type 1 (SCA1) transgenic mouse model. Specifically, it regulates GSK3β dephosphorylation through inhibiting PP2A holoenzyme, which is involved in GSK3β dephosphorylation. However, *ANP32a* overexpression restores GSK3β phosphorylation at Ser9 and alterations in neurite morphology, such as neurite length changes, induced by mutant ataxin-1 [39]. These results suggest that *ANP32a* plays an important role in the development and differentiation of the nervous system. Furthermore, if *ANP32a* is overexpressed in cortical neurons, it remains in the nucleus to protect neurons against N-methyl-d-aspartate acid (NMDA) toxicity [37]. Through the present work, we add to this cumulative literature by showing that *ANP32a* contributes to the regeneration of spinal cord neurons in zebrafish embryos after SCI.

In contrast, *ANP32a* may also negatively affect neural cells during human disease. For example, the level of *ANP32a* is increased in the brain of Alzheimer’s patients, where it acts as an inhibitor of histone acetyltransferase complex, a multi-protein complex that potently inhibits specific histone acetyltransferases, causing transcriptional repression and leading to negative regulation of protein expression [40,57]. Upregulated *ANP32a* also increases Tau phosphorylation, impairing the microtubule network and neurite outgrowth in Alzheimer’s disease [39]. However, after knockdown of *ANP32a* in neuronal cell lines, neurite outgrowth was promoted. Moreover, the outgrowth of primary neurons from *ANP32a*-null mice was enhanced compared with that in wildtype mice [58]. These data suggest that *ANP32a* may inhibit neurite outgrowth and neural differentiation.

### 3.3. Cell Signaling Pathways with Possible Involvement in Promoting Cell Proliferation Mediated by ANP32a

*ANP32a* may also induce gene transcription. For example, *ANP32a* could be an oncogene that promotes glioma cell proliferation by regulating the Akt/p27/stathmin pathway [52]. *ANP32a* can induce IFN-stimulated genes through interacting with signal transducer and activator of transcription 1 (STAT1) and STAT2 [28]. Furthermore, *ANP32a* could positively regulate the HMGA1/STAT3 pathway through interaction with *HMGA1* mRNA, followed by activation of STAT3 [36]. The STAT1 pathway is one of the most important transducing signals in response to cytokine induction and inflammatory response after SCI [59,60,61,62]. The STAT3 protein is significantly increased after peripheral nerve injury [63]. The JAK/STAT3 pathway is activated in spinal cord microglia after peripheral nerve injury [64]. Okada et al. (2006) [65] also revealed that STAT3 is a key regulator of reactive astrocytes in the repair of injured tissue and the recovery of motor function after SCI. Collectively, these findings point to zebrafish *ANP32a* as a plausible positive regulator of neural regeneration through activation of STAT signaling in radial glial cells and motor neurons during regeneration of SCI zebrafish embryos.

Nevertheless, it is already known that reactive oxygen species and oxidative stress play a significant role in the pathophysiology of SCI. After SCI, the neurons and glial cells are particularly prone to oxidative and electrophilic stress owing to the high content of polyunsaturated fatty acids, high rate of oxidative metabolic activity, intense production of reactive oxygen metabolites and relatively low antioxidant capacity. In fact, oxidative stress is considered a hallmark of the secondary phase of SCI [66,67]. Therefore, alleviating oxidative stress may be an effective strategy for therapeutic interventions for SCI. Cornelis et al. (2018) [68] reported that *ANP32a* promotes transcription of Ataxia Telangiectasia Mutated (ATM), which is an essential checkpoint kinase that signals DNA double-strand breaks in eukaryotes as well as a key regulator of cellular oxidative defense. They also reported that antioxidant therapy protects *ANP32a*-deficient mice from developing osteoarthritis and osteopenia, in addition to rescuing neurological defects. Thus, *ANP32a* might be a positive regulator of neural regeneration through the reduction of oxidative stress. When considered collectively, many avenues of further investigation remain open. For instance, we do not understand which signaling pathway(s) *ANP32a* takes in its involvement in neural regeneration. We also do not understand what triggers *ANP32a* to play active or repressive roles in different physiological processes and in different cell types. Thus, deciphering *ANP32a* signaling that promotes endogenous spinal cord regeneration in zebrafish might reveal its regenerative processes in mammalian contexts.

### 3.4. Specific Phenotypes Induced by MO Knockdown in Zebrafish Embryos

In this study, we employed loss-of-function to determine gene function, i.e., whether the phenotypes of MO-injected embryos resulted from specific induction or the toxicity of injected MO. As shown in Figure 2, Figure 3 and Figure 4 in this study, we first performed a dose-responsive assay of *Anp32a*-MO (Appendix A). We examined 1, 1.5, 1.75, 2 and 4 ng prior to each 2.3 nl injection for the knockdown experiment. The results demonstrated that the phenotypes induced by injected MO were dose-dependent. Among them, 2 ng was the most suitable concentration for further study. Second, we injected wobble-*Anp32a*-flag mRNA to see whether the phenotypes of MO-injected embryos had been specifically rescued by co-injection of *Anp32a*-flag mRNA in a dose-dependent manner (Appendix A). The results showed that the rescue effect of spinal cord regeneration was increased when the concentration of *Anp32a*-flag mRNA was increased from 200 to 400 pg. Additionally, the minimal concentration of *Anp32a*-flag mRNA that could specifically rescue MO-induced defects was 200 pg. This line of evidence suggests that the defective phenotypes induced by the concentration of *Anp32a*-MO used to knock down *Anp32a* in this study did not result from the onset of toxicity.

Although the use of mutants would be more convincing, no zebrafish *Anp32a* mutant is currently available. Despite some concerns over using the MO-base gene knockdown approach, Morpholino oligomers remain an essential tool to transiently inhibit gene expression in the zebrafish model, as attested by the numerous MO-related papers published by our lab [69,70,71,72]. As noted above, we employed MO-knockdown combined with rescue through *Anp32a* mRNA and Western blot analysis to demonstrate that the defective phenotype of spinal cord regeneration induced by *Anp32a*-MO was specific.

### 3.5. Conclusions

In conclusion, we demonstrated that overexpression of *Anp32a* significantly improved the regeneration of motor neurons and swimming capability of larvae derived from SCI embryos, while knockdown of *Anp32a* displayed a decidedly negative effect on those parameters. Moreover, immunofluorescence staining, when combined with FACS, demonstrated that *ANP32a* could promote cell proliferation. Specifically, the induction of proliferating radial glial cells and motor neurons facilitated the regeneration of motor neurons in the SCI embryos of zebrafish. This line of evidence strongly supports a close association between *ANP32a* and proliferating neural cells in zebrafish SCI embryos. Therefore, we speculate that the mechanism underlying the association between *ANP32a* and neuronal regeneration after SCI likely involves the activation of signal transducer and activator of transcription (STAT) signaling in motor neurons and radial glial cells during regeneration.

## 4. Materials and Methods

### 4.1. Zebrafish

Zebrafish wild-type AB strain and transgenic lines, including *Tg(gfap:GFP)* [73], *Tg(mnx1:GFP)* [74] and *Tg(mnx1:TagRFP)* [75], were cultured and maintained using standard procedures [76]. Embryo medium containing 0.003% 1-phenyl-2-thiourea (Sigma; P7629, St. Louis, MO, USA) was used at 12 hpf to reduce pigmentation. Fluorescence was visualized with a fluorescent stereomicroscope (Leica Microsystems, Wetzlar, Germany) and a confocal spectral microscope (Leica Microsystems, Wetzlar, Germany).

### 4.2. Mechanical Crush Injury to Induce Spinal Cord Injury of Zebrafish Larvae

We followed the protocol described by Zeng et al. [21]. Briefly, embryos at 48 hpf were immersed in embryo medium containing 0.02% tricaine (Sigma; E10521, St. Louis, MO, USA) for anesthesia. A tweezer (Dumont No.5, São Paulo, Brasil) was employed to achieve a spinal cord crush lesion between the 17th and 20th somite, located above the anus. These SCI embryos were placed in fresh medium containing 0.003% 1-phenyl-2-thiourea for study. Apart from precise definition of the SCI site, we also tried to increase the sample size in order to reduce individual differences.

To understand neuronal regeneration after SCI, we used “SCI embryos” as a control instead of “control no SCI embryos”, because the results were based on the experiments summarized in Figure 3 and Figure 5. Therefore, we investigated differences between either SCI- and SCI+ mRNA groups or SCI- and SCI + MO groups in order to determine the effect of overexpression or knockdown of mRNA on the neuronal regeneration of SCI embryos on a comparative basis. In other words, the essential control group in this respect was SCI embryos as a control, as opposed to “control no SCI embryos” group.

### 4.3. Confocal Microscopy and Image Processing

Fluorescence signals were captured using a Zeiss (Oberkochen, Germany) confocal microscope (LSM 780) and Olympus U-HGLGPS. Images were analyzed using ImageJ and ZEN2009 Light Edition. For time-lapse imaging of embryos, we embedded embryos in a 2–3% methylcellulose gel (Sigma; M5012, St. Louis, MO, USA) for image acquisition and replaced the culture media every 3 or 6 h.

### 4.4. Microinjection and mRNA Synthesis

We followed the procedures previously described by Lee et al. [77,78], except that plasmids pCS2-*Anp32a*-flag and pCS2-wobble-*Anp32a*-flag were linearized by NotI (NEB). The capped *Anp32a-flag* and wobble-*Anp32a*-flag mRNA were synthesized using the SP6 Message Machine Kit (Ambion, Austin, TX, USA) and diluted to a working concentration of 88 ng µL^−1^ prior to each 2.3 nl injection. The MO was purchased from Gene Tools (USA) and prepared according to the protocol published by Gene Tools, which calls for dilution to a working concentration of 2 ng prior to each 2.3 nl injection. The sequence of MO used in this study was as follows: *Anp32a*-MO: 5′-CTCTTTTTCATATCCATCTCTGTGA-3′.

To reduce biased data from the transgenic studies during each experiment, we first chose normally developed embryos to perform gene transfer. After transgenesis, we discarded transgenic embryos exhibiting bizarre appearance, and then randomized to select the samples, ruling out samples displaying extreme values. Transgenic embryos that exhibited abnormal expression patterns, compared to control, were also deleted. Each trial resulted in a random selection that was repeated at least three to five times to obtain a single average. Then, the final data presented in the manuscript were averaged from three independent experiments, followed by the appropriate statistical analysis. Last, apart from precise definition of the SCI site, we also tried to increase the sample size in order to reduce individual differences. For example, in Figure 5B,C, 11 SCI embryos were analyzed in the 18-hpi group and 11 in the *Anp32a* mRNA plus SCI group, while 10 were analyzed in the 24-hpi group and 10 in the *Anp32a* mRNA plus SCI group.

### 4.5. FACS

A FACSAria cell sorting system (BD Biosciences, FACSVerse™, San Jose, CA, USA) and SH800 cell sorter (SONY, Tokyo, Japan) were used to perform FACS in order to sort out different fluorescent signal-expressing cells under sterilized conditions, according to the protocols described by Dobson et al. [79]. Using antibodies against Olig2 (1:200; Invitrogen; PA5-85734, Carlsbad, CA, USA) in FACS, we followed the protocol described by Zeng et al. [80] with some modifications. The *Anp32a* mRNA was microinjected into the one-cell stage of embryos from wild-type or *Tg(gfap:GFP)*, followed by SCI at 48 hpf and deheading. Cells were suspended and stained with antibodies against Olig2 and DAPI (1:1000; Sigma; D8417, St. Louis, MO, USA) at 24 hpi. We randomly collected 100 SCI embryos, suspended cells from deheaded embryos, fixed the samples and performed cell immunostaining using Olig2 antibodies labeled with RFP or DAPI staining. After analyzing the cell population, the percentage of GFP/RFP-positive cells expressing DAPI signals was determined.

Embryonic cells from wild-type zebrafish at 72-hpf served as negative control in order to determine which strength of fluorescence was considered negative. Then, we applied RFP-stained cells (RFP) or GFP-expressing cells from transgenic line (*Tg(gfap:GFP)*) to determine which strength of fluorescence was considered positive.

### 4.6. Immunohistochemistry

Immunostaining was performed as previously described [21]. Primary antibodies were used as follows: Monoclonal anti-phospho-Histone H3 (Ser10) (1:200; Millipore; H0412, Burlington, MA, USA), Polyclonal anti-Olig2 (1:200; Invitrogen; PA5-85734) and goat anti-rabbit IgG- Rhodamine (1:250; Sigma; AP132R, St. Louis, MO, USA). For BrdU staining, embryos at 48 hpf were microinjected with 4.6 nl of 50 mM BrdU (Sigma; B5002) into embryo ventricle brain, followed by SCI, and incubated at 18 and 24 hpi for further analysis. Brain ventricle injections were performed as described by Gutzman and Sive [81].

### 4.7. Western Blot Analysis

Total proteins extracted from embryos were analyzed on a 15% SDS-PAGE, followed by Western blot analysis, according to the procedures described by Lee et al. [77], except that the antibodies against *ANP32a* (1:1000; Aviva Systems Biology; ARP40204_T100, San Diego, CA, USA), α-tubulin (1:5000; Sigma; RRID:AB_477579, St. Louis, MO, USA), Flag (1:1000; Abcam; RRID:AB_446355, Cambridge, UK), goat anti-rabbit-HRP (1:10,000; Cell Signaling; RRID:AB_2099233, Frankfurt, Germany) and goat anti-mouse-HRP (1:10,000; Abcam; RRID:AB_955439, Cambridge, UK) were used. Ninety micrograms of extracts were loaded to analyze the specific proteins from zebrafish embryos. The signal intensity of the band was analyzed using ImageJ software.

To reduce biased data, we only took into account the intensity of *ANP32a* when the expressional intensity of the internal control, alpha-tubulin, loaded into each well on gel was almost identical on the same blot paper.

### 4.8. Swimming Capability Assay

The procedures for the swimming capability assay followed those of Zeng et al. [21] with some modifications. At 24 hpi, each larva was kept in a 3 cm plate. After touching the head, a high-speed camera (Photron, FASTCAM SA1.1, Tokyo, Japan) was used to record swimming performance. Each individual larva was recorded three times, and Fiji ImageJ was used to analyze and quantify the larval swimming route. The route sampled in the video was converted into a set of coordinates. Then, each coordinate was converted into swimming distance in mm using Excel.

### 4.9. Statistical Analysis

Unless otherwise indicated, each experiment was performed at least three times. Animals were randomly assigned to different experimental groups, but no formal method of randomization was used. We used one-way ANOVA, followed by Tukey’s multiple comparison test, or student *t* test for comparisons. Significance was determined at *p* value as indicated in the figure legends. Statistical analyses were performed with Microsoft Excel or GraphPad Prism v9.

## Figures and Tables

**Figure 1 ijms-23-15921-f001:**
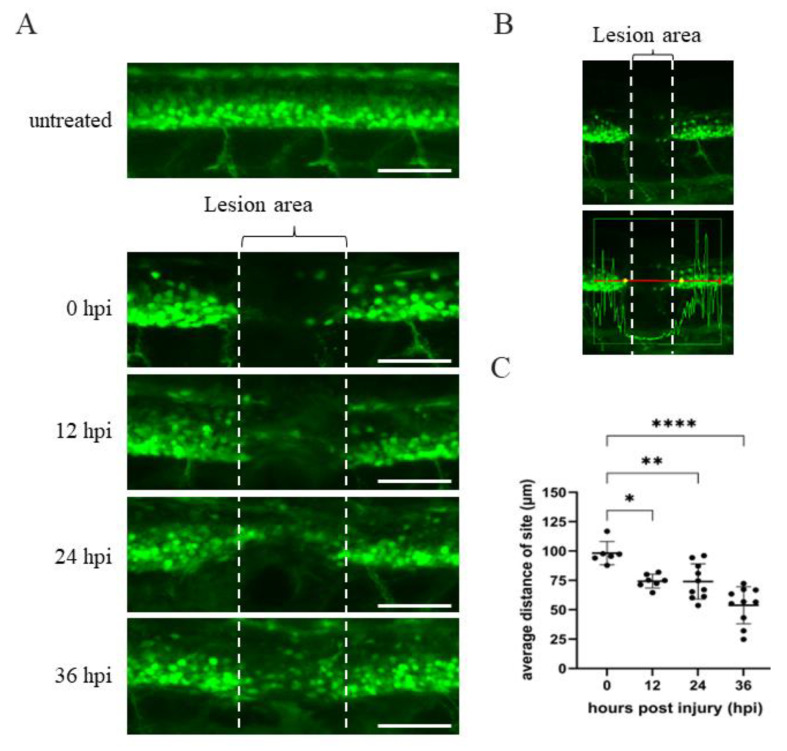
Time course of spinal cord regeneration of *Tg(mnx1:GFP)* transgenic zebrafish larvae after SCI. (**A**) Spinal cord located at the 17–20th segment of *Tg(mnx1:GFP)* embryos at 48 hpf was crushed, causing SCI. Complete loss of motoneuronal connections in the lesion area around the SCI (indicated by two broken lines) occurred at 0 hpi. Then, spinal cord regeneration from 12 to 36 hpi was shown, compared to uninjured spinal cord. (**B**) Image of SCI-*Tg(mnx1:GFP)* larvae at 0 hpi used to quantify the length of lesion site (in μm) using Zen 2010b Sp1 software. The box in the lower panel shows a red line 300 μm in length. This indicates the detection area with fluorescent cells located at either side of the SCI site, as marked by the vertical white broken line. While the GFP signal within the box could be detected, its intensity dropped at the SCI site. Indeed, the yellow dots mark the place where the strength of the green fluorescence signal dropped off at both sides of the SCI. The length of yellow dots relative to that of the vertical broken white line was measured. Observed by the naked eye, the length of yellow dots was longer than that of SCI site (white broken line). (**C**) Statistical analysis of the distance (in μm) between the two yellow dots versus temporal regenerative response (in hpi) in SCI-*Tg(mnx1:GFP)* embryos. Each dataset was obtained from three independent experiments and represented as mean ± SEM. One-way ANOVA was used to perform statistical analysis (* *p*  <  0.05, ** *p*  <  0.01, **** *p*  <  0.0001; ns: not significant). Scale bar: 100 μm.

**Figure 2 ijms-23-15921-f002:**
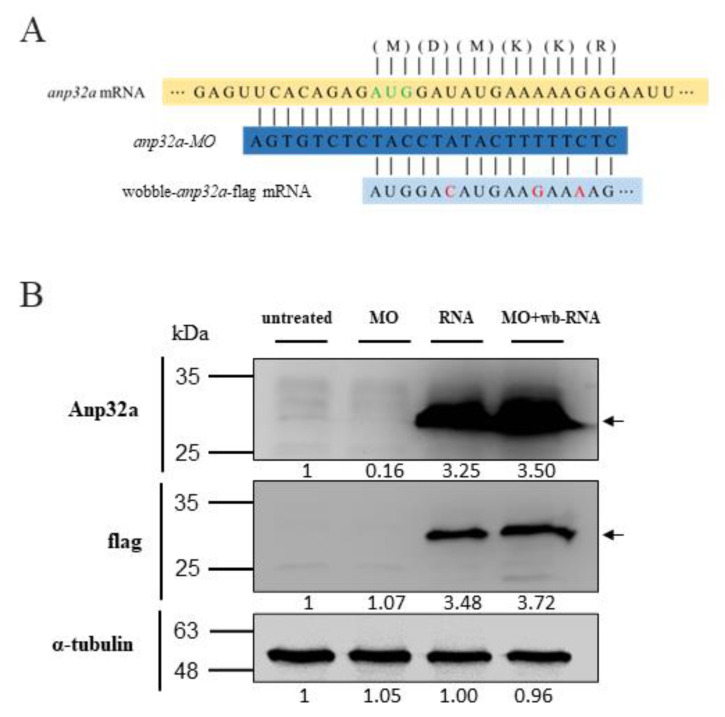
Western blot analysis of *ANP32a* in zebrafish embryos undergoing either knockdown or overexpression of *Anp32a*. (**A**) Diagrams were used to depict base-pairings among *Anp32a* mRNA, antisense morpholino oligonucleotides (MO) of *Anp32a* (*Anp32a*-MO) and wobble nucleotides of *Anp32a*-flag mRNA (wobble-*Anp32a*-flag mRNA). (**B**) Western blot analysis. Zebrafish embryos at one-cell stage were microinjected with *Anp32a*-MO (MO), wobble-*Anp32a*-flag mRNA (wb-RNA) and MO plus wobble mRNA (MO + wb-RNA) as indicated, followed by extraction of total embryonic proteins at 48 hpf. Protein levels of endogenous *ANP32a* and exogenous *ANP32a*-flag were detected using antibodies against *ANP32a* and reporter protein flag. α-tubulin served as an internal control. Protein levels relative to each internal control are presented below each lane.

**Figure 3 ijms-23-15921-f003:**
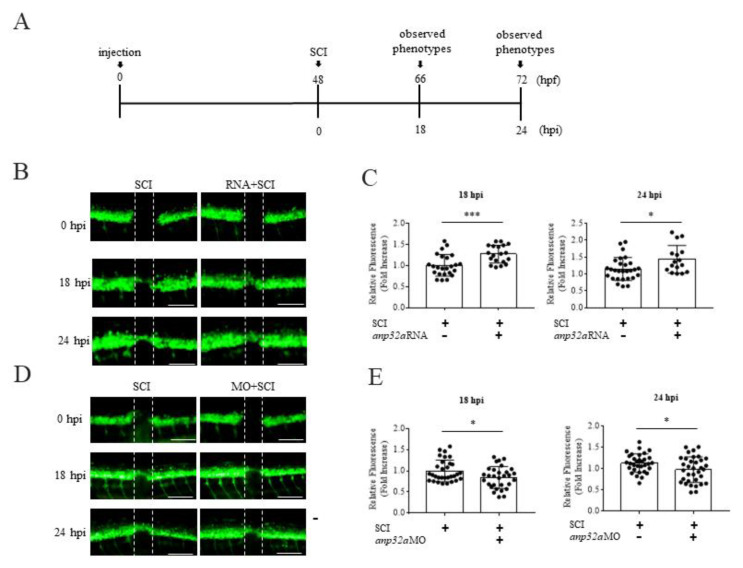
*ANP32a* is required for the neuronal regeneration of zebrafish larvae after SCI. (**A**) The experimental flowchart showing the time course of microinjection of either *Anp32a* mRNA or *Anp32a* morpholinos, SCI performance and observation of spinal cord regeneration. Embryos at the one-cell stage were injected with either wobble-*Anp32a*-flag mRNA or antisense morpholino oligonucleotides (MO) of *Anp32a* and SCI at 48 hpf, followed by analysis of the lesion gap at 0, 18 and 24 hpi. (**B**) Time course image of spinal cord regeneration in control embryos and *Anp32a*-overexpressed embryos at 0, 18 and 24 hpi. (**C**) Quantification of temporal regenerative response in SCI-control and wobble-*Anp32a*-flag mRNA-injected embryos at 18 hpi (left; control: *n* = 25; RNA + SCI: *n* = 19) and 24 hpi (right; control: *n* = 28; RNA + SCI: *n* = 16). (**D**) Time course image of spinal cord regeneration in control embryos and *Anp32a*-morphants at 0, 18 and 24 hpi. (**E**) Quantification of temporal regenerative response in SCI control embryos and *Anp32a*-morphants at 18 hpi (left; control: *n* = 31; MO + SCI: *n* = 30) and 24 hpi (right; control: *n* = 32; MO + SCI: *n* = 34). Scale bar: 100 μm; The Student’s *t*-test was used to perform statistical analysis (*, *p* < 0.05; *p* < 0.01; ***, *p* < 0.001; error bars indicate mean ± SD).

**Figure 4 ijms-23-15921-f004:**
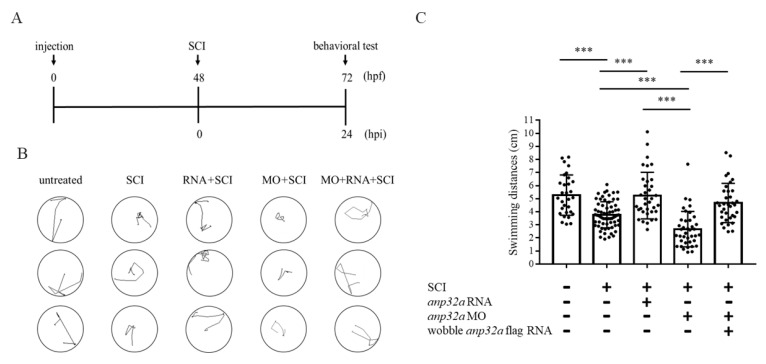
Defective swimming activity was partially recovered in the SCI larvae with *Anp32a* overexpression. (**A**) Experimental flowchart showing the time course of microinjection of either *Anp32a* mRNA (RNA) or *Anp32a* morpholinos (MO), SCI performance and behavior test. Embryos at the one-cell stage were microinjected with either wobble-*Anp32a*-flag mRNA or *Anp32a*-MO. Following SCI at 48 hpf, we recorded the swimming trajectory route at 24 hpi. (**B**) To carry out gain-of-function and loss-of function studies, the swimming trajectory route of each fish from three groups was examined, including untreated control (neither injection nor SCI), SCI only (SCI), both wobble-*Anp32a*-flag mRNA injection and SCI (RNA + SCI), injection of *Anp32a*-MO and SCI (MO + SCI), and co-injected *Anp32a*-MO with wobble-*Anp32a*-flag mRNA injection and SCI (MO + RNA + SCI). (**C**) Quantification and statistical analysis. Swimming distances (cm) were calculated based on the recorded trajectory route shown in (**B**). The data in each group were averaged from all examined larvae (untreated control: *n* = 30; SCI: *n* = 66; RNA + SCI: *n* = 35; MO + SCI: *n* = 39; MO + RNA + SCI: *n* = 35). One-way ANOVA, followed by Tukey’s multiple comparison test, was used to perform statistical analysis (***, *p* < 0.001; error bars indicate mean ± SD).

**Figure 5 ijms-23-15921-f005:**
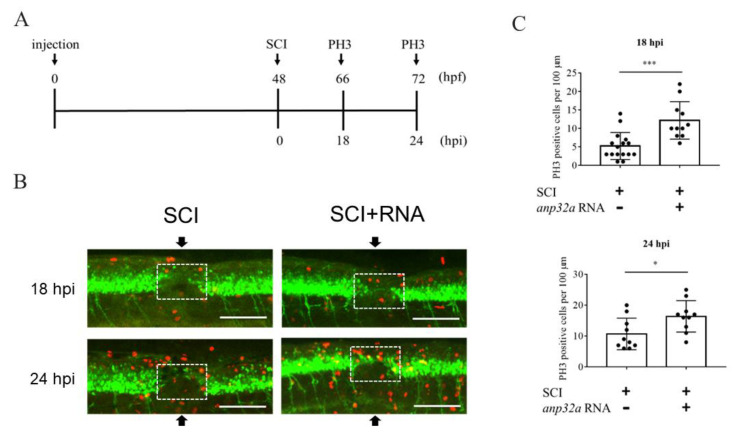
Overexpression of *Anp32a* could promote cell proliferation in SCI embryos during recovery. (**A**) The experimental flowchart. *Anp32a* mRNA was injected into embryos from zebrafish transgenic line *Tg(mnx1:GFP)* at the one-cell stage, followed by SCI at 48 hpf and staining with anti-PH3 at 18 and 24 hpi. (**B**) Proliferation patterns around the lesion site of embryos treated as indicated at 18 and 24 hpi were presented by immunofluorescence through antibody against phosphorylated histone H3 (PH3; Red) and visualized by laser confocal microscopy. Arrow indicates the SCI site, while white dotted box indicates the calculation area. (**C**) Statistical count of PH3-positive cells between SCI-*Anp32a*-overexpressed embryos (*n* = 11 at 18 hpi; *n* = 10 at 24 hpi) and SCI embryos without overexpression of *Anp32a* (*n* = 11 at 18 hpi; *n* = 10 at 24 hpi). A) Scale bar: 100 μm. The Student’s *t*-test was used to perform statistical analysis (*, *p* < 0.05; *p* < 0.01; ***, *p* < 0.001; error bars indicate mean ± SD).

**Figure 6 ijms-23-15921-f006:**
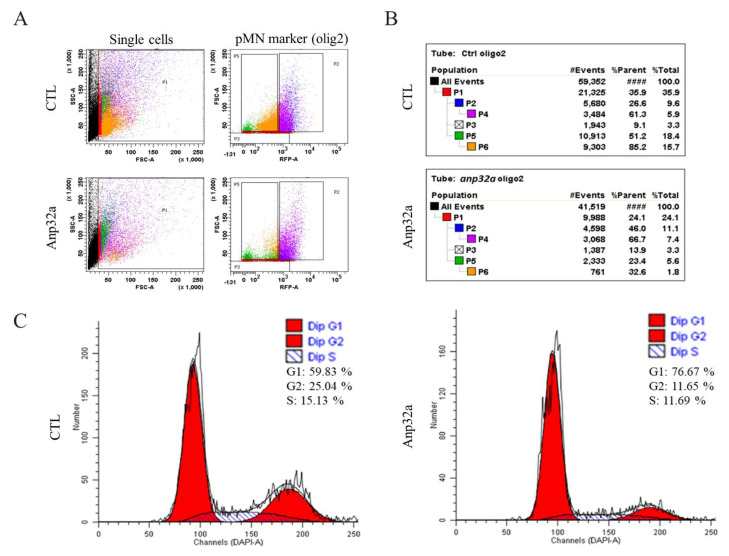
The overexpression of *Anp32a* in SCI embryos of zebrafish could induce an increase in the cell number of motor neuron progenitor cells (pMN) around the SCI site. Fluorescence-activated cell sorting (FACS) and a cell cycle analysis of pMN in *Anp32a*-overexpressing SCI embryos of zebrafish were performed. The *Anp32a* mRNA was microinjected into the one-cell stage of embryos from wild-type zebrafish, followed by SCI at 48 hpf. After deheading, suspended cells were stained with antibodies against Olig2 and DAPI at 24 hpi. (**A**) Scatter profiles of *Anp32a*-overexpressed SCI-zebrafish embryonic cells. All analytic cells were gated at P1; RFP-labelled Olig2-positive pMN cells were gated at P2; DAPI-labeled-positive cells were gated at P4; Olig2-negative cells were gated at P5. (**B**) Typical FACS dataset. The number shown in the right corner represents the percentage of cells at each gate relative to the total number of examined cells in each group. (**C**) Cell cycle analysis from DAPI-labeled-positive cells shown on P4.

**Figure 7 ijms-23-15921-f007:**
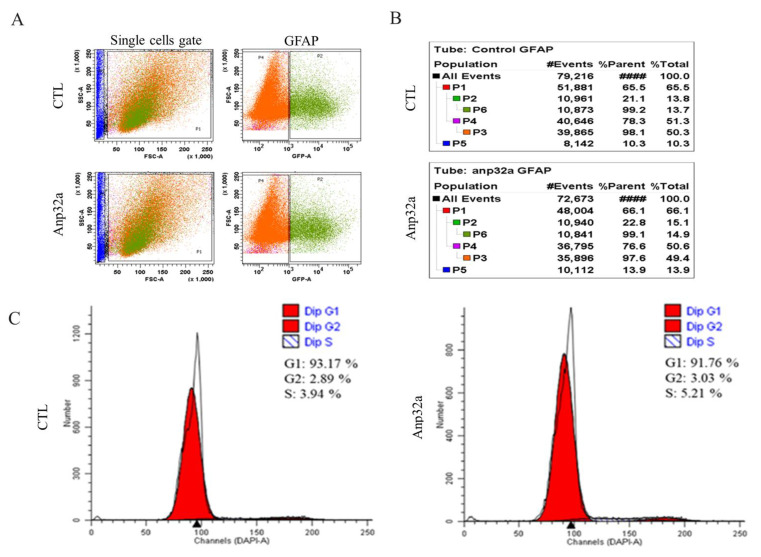
Overexpression of *Anp32a* in SCI embryos of zebrafish could induce the active proliferation of radial glial cells around the SCI site. Fluorescence-activated cell sorting (FACS) and a cell cycle analysis of radial glial cells in *Anp32a*-overexpressing SCI embryos of zebrafish were performed. *Anp32a* mRNA was microinjected into one-cell stage embryos from *Tg(gfap:GFP)* zebrafish, followed by SCI at 48 hpf and DAPI staining at 24 hpi. (**A**) Scatter profiles of *Anp32a*-overexpressing *Tg(gfap:GFP)* SCI-embryonic cells. All analytic cells were gated at P1; GFP-positive cells were gated at P2; DAPI-labeled-positive cells were gated at P6; GFP-negative cells were gated at P4. (**B**) Typical FACS dataset. The number shown in the right corner is the percentage of proliferated cells at each gate relative to the total examined cells in each group. (**C**) Cell cycle analysis from DAPI-labeled-positive cells shown on P6.

**Figure 8 ijms-23-15921-f008:**
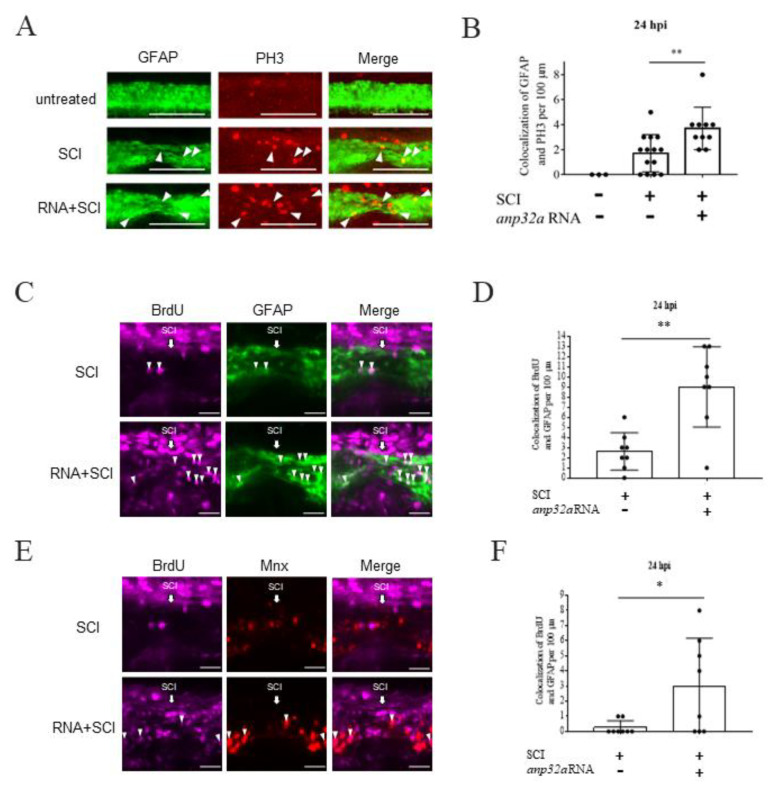
Proliferation and cell number of radial glial cells and motor neurons were increased in the *Anp32a*-overexpression SCI embryos. The *Anp32a* mRNA was microinjected into the one-cell stage of embryos from zebrafish transgenic lines (**A**,**B**) *Tg(gfap:GFP)* and (**C**–**F**) the crossed strain of *Tg(gfap:GFP)* and *Tg(mnx1:TagRFP)*, followed by BrdU injection, SCI at 48 hpf, and finally, staining with PH3 at 24 hpi. (**A**) Proliferation patterns at the lesion site of *Tg(gfap:GFP)* embryos treated as indicated at 24 hpi were displayed by immunofluorescence through reaction with antibodies against phosphorylated histone H3 (PH3; Red) and observation under laser confocal microscopy. (**B**) Statistical analysis showed a significantly increased number of PH3-positive cells in *Anp32a*-overexpressed SCI embryos (*n* = 10) compared to untreated embryos (*n* = 3) and SCI-control embryos (*n* = 14) at 24 hpi. One-way ANOVA followed by Tukey’s multiple comparison test was used to perform statistical analysis (*, *p* < 0.05; ** *p* < 0.01; error bars indicate mean ± SD). (**C**) Proliferation patterns at the lesion site of transgenic cross strain *Tg(gfap:GFP)* X *Tg(mnx1:TagRFP)* embryos treated as indicated at 24 hpi showing GFAP-positive cells (green) colocalized with BrdU-positive cells (purple) around the lesion area. (**D**) Statistical analysis of GFAP +/BrdU+ colocalized cells in SCI-control and *Anp32a*-overexpressing SCI embryos. (**E**) Proliferation patterns at the lesion site of *Tg(gfap:GFP)* X *Tg(mnx1:TagRFP)* embryos treated as indicated at 24 hpi showing red-labelled motor neurons colocalized with BrdU-positive cells (purple) around the lesion area. The specimens in Figure 8C, F originated from the same embryos. (**F**) Statistical analysis of motor neuron +/BrdU + colocalized cells in SCI-control and *Anp32a*-overexpression SCI embryos. Scale bar: 100 μm; (**D**,**F**) Student’s *t*-test was used to perform statistical analysis (* *p* < 0.05; ** *p* < 0.01; error bars indicate mean ± SD).

## Data Availability

The data that support the findings of this study are available from the corresponding authors upon request.

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
