# Peer review of "Anp32a Promotes Neuronal Regeneration after Spinal Cord Injury of Zebrafish Embryos"

_ijms, 2022, doi:10.3390/ijms232415921_

Round 1

Reviewer 1 Report

This manuscript describes the effects of knockdown and overexpression of the gene anp32a during spinal cord regeneration. The authors use overexpression of the gene for gain of function and one morpholino for loss of function. They show that the lesion area is more quickly repopulated by neurons and axons when anp32a mRNA is increased and more slowly when translation is impaired. Furthermore, they show that the proliferation of olig2 positive cells is slightly increased. These could be motor neuron or oligodendrocyte progenitors, which is not conclusively discussed. 

There is some novelty in these results, however, it is a shame that there are multiple elements of a story missing. First and foremost, it is unclear why the authors would have wanted to look at anp32a. This also doesn't become clear in the discussions, where all previously proposed roles of anp32a are mentioned, but no coherent hypothesis is presented that their data would support. 

There are a couple of instances of reuse of images. In figure 3, the two panels of the 0 hour time points in 3B and 3D are identical, but the experimental treatment for the right columns are different. 

Across the manuscript are instances of unclear terminology and interpretation at odds with the literature. I detail these below:

“rapid neuronal regeneration“ this is a logical error. The glial scar mainly impairs axonal regeneration. The regeneration of neurons is a (partly) separate process. 

Re: Fig 1 and SCI model – how did the authors choose the time point of the injury? Larval spinal cord injury is well characterised by several groups, especially the Dorsky and Becker labs, who injure at 3 dpf and later. At 48 hours, developmental neurogenesis is still ongoing, so it’ll be difficult to distinguish between specific developmental and regenerative processes.

Olig2 does not only label motor neuron progenitors, but also oligocytes and OPCs. There is a developmental switch from motorneurogenesis to oligodendrogenesis that starts a few hours before they lesion (see works of the Appel lab). 

Migration of mnx1:gfp positive neurons into a spinal injury has been described, how do the authors distinguish between axons and neurons if they just look at fluorescence levels in the injury site? There is imprecision in the language throughout between neuronal and axonal regeneration. Furthermore, the authors do not formally show that the neurons in the injury site are new. EdU experiments would have to be performed to state this with confidence, especially since it was recently reported that pre-existing neurons can migrate into the injury site in early phases of regeneration (Vandestadt et al Dev Cell 2021).

In the analyses presented in Figure 4, the wrong statistics have been applied, since the authors are using the same groups for multiple comparisons. An ANOVA with a posthoc test would be more appropriate. This is mentioned in the statistics, but not in the figure legends.

Methods to reduce bias have been insufficiently discussed (apart from randomisation).

The specimens in Figure 8 C and F are the same fish - this should be pointed out clearly.

minor:

„lower vertebrate“ is an inaccurate designation for fishes and amphibians. “Many anamniotes” or “many fishes and amphibians” can be used instead. The “many” is needed because post-metamorphic and adult frogs cannot regenerate spinal cord. Same for “higher vertebrate” used further down the text. 

In the sentence “Furthermore, no permanent glial scar formation does not” there is a double negative

“the absence of immune rejection during transplantation makes Zebrafish the dominant model system for real-time screening of transplanted cells harboring genes vital for successful regeneration after SCI.“ Ma et al did not transplant any cells and I am personally only aware of the present authors who have transplanted cells into the zebrafish spinal cord after injury. It is therefore puzzling that the authors would describe the zebrafish as the “dominant model system” for such approaches.

“The spinal cord between 17-20 th somite developed at 48 hpf, which is located above the anus of embryos from Tg(gfap:GFP), was transected by tweezer.“ It is not clear what is meant here – can the author rephrase.

“treated with SCI” – spinal cord injury is not a treatment. 

Proliferation studies: it is not clear that neural cells proliferate, clearly this would not be post-mitotic neurons (as labelled by the line in figure 5).  A spinal cord injury leads to the influx of a significant number of non-neural cells into the injury site (immune cells, fibroblasts, keratinocytes) – which cells proliferate?

Reference 45 does not use an antibody against olig2 – how was the antibody used in FACS experiments validated?

Author Response

Reviewer 1

This manuscript describes the effects of knockdown and overexpression of the gene anp32a during spinal cord regeneration. The authors use overexpression of the gene for gain of function and one morpholino for loss of function. They show that the lesion area is more quickly repopulated by neurons and axons when anp32a mRNA is increased and more slowly when translation is impaired. Furthermore, they show that the proliferation of olig2 positive cells is slightly increased. These could be motor neuron or oligodendrocyte progenitors, which is not conclusively discussed.  

  1. There is some novelty in these results, however, it is a shame that there are multiple elements of a story missing. First and foremost, it is unclear why the authors would have wanted to look at anp32a. This also doesn't become clear in the discussions, where all previously proposed roles of anp32a are mentioned, but no coherent hypothesis is presented that their data would support. 

Authors’ response to 1:

In response to your suggestion, we added our rationale for studying anp32a in the revised Introduction. In summary, we previously used transcriptomic analysis to screen key regulatory genes involved in neuronal regeneration after spinal cord injury (SCI) of zebrafish embryos (Zeng et al., 2021). Among 3911 candidate genes, they categorized all of them into four groups based on Gene Ontology. In our study, we focused on candidate genes in the regeneration/development-related gene group and selected runx3, elav11A, crabp1a, bace1, inpp5e and anp32a. Among these candidates analyzed in preliminary studies, only knockdown of either inpp5e or anp32a could significantly impair swimming ability in SCI-embryos (see attached figures below). Furthermore, the impairment induced by anp32a-MO was more serious than that induced by inpp5e-MO. Therefore, we chose anp32a for further study. We added this detailed information in the Introduction (Please see lines 36 to 45 of page 2).

Additionally, we explained in the Discussion how our data were consistent with the hypothesis that anp32a is related to the regeneration of SCI-embryos of zebrafish, as follows: “In this study, we demonstrated that overexpression of anp32a significantly improved the regeneration of motor neurons and swimming capability of larvae derived from SCI-embryos, while knockdown of anp32a displayed a decidedly negative effect on those parameters. Moreover, immunofluorescence staining combined with fluorescence-activated cell sorting (FACS) demonstrated that Anp32a could enhance cell proliferation. Specifically, the induction of proliferating radial glial cells and motor neuron progenitor cells facilitated the regeneration of motor neurons in the SCI-embryos of zebrafish. This line of evidence strongly supports a close association between Anp32a and proliferating neural cells in zebrafish SCI-embryos. Therefore, we speculate that the mechanism underlying the involvement of Anp32a in neuronal regeneration after SCI might be through the activation of signal transducer and activator of transcription (STAT) signaling in motor neurons progenitors and radial glial cells during regeneration.” (Please see the last paragraph of Discussion section, lines 16 to 28 of page 18)

  1. There are a couple of instances of reuse of images. In figure 3, the two panels of the 0 hour time points in 3B and 3D are identical, but the experimental treatment for the right columns are different. 

Authors’ response to 2:

Thank you for pointing out this mistake. In the revised manuscript, we changed the 0-hour time point image in 3B, as shown below.

Across the manuscript are instances of unclear terminology and interpretation at odds with the literature. I detail these below:

  1. “rapid neuronal regeneration“ this is a logical error. The glial scar mainly impairs axonal regeneration. The regeneration of neurons is a (partly) separate process. 

Authors’ response to 3:

Thank you for pointing out this mistake.  We revised, as follows: “Furthermore, glial scar formation of zebrafish is restricted after SCI, resulting in favoring rapid axonal regeneration and improvement of neuronal regeneration of SCI- zebrafish.” Please see lines 20 to 22 of page 2.

  1. Re: Fig 1 and SCI model – how did the authors choose the time point of the injury? Larval spinal cord injury is well characterised by several groups, especially the Dorsky and Becker labs, who injure at 3 dpf and later. At 48 hours, developmental neurogenesis is still ongoing, so it’ll be difficult to distinguish between specific developmental and regenerative processes.

Authors’ response to 4:

We chose to study embryos at 48 hpf because exogenous injection of either mRNA or MO is barely retainable in embryos after 72 hpf. Therefore, gain- and loss-of function experiments performed in this study would have been difficult to perform in embryos at 72 hpf following SCI. While oligodendrogenesis from neural progenitors continues for weeks (Park et al., 2007), embryonic motor neuron generation from the pMN domain largely ceases between 48 and 51 hpf (Reimer et al., 2013). Moreover, Ohnmacht et al. (2016) demonstrated that local lesion-induced pMN signals lead to motor neuron regeneration using Hb9:GFP (also known as mnx1:GFP) transgenic zebrafish, suggesting that GFP-expressing cells of Tg(mnx1:GFP) embryos at the lesion site are the newly formed motor neurons. In this study, we observed that RFP-expressing cells of Tg(mnx1:RFP) SCI-embryos at the lesion site were co-localized with BrdU signal (Figure 8), suggesting that RFP+/BrdU+ cells were newly formed motor neurons during spinal cord regeneration after SCI. This result was consistent with that reported by Ohnmacht et al. (2016).

Although developmental neurogenesis is still ongoing at 48 hpf, as you mentioned, our data indicated that the recovery of spinal cord and swimming capability was significantly improved or retarded in SCI larvae treated with gain- and loss-of-function, respectively, at 48 hpf, compared to control developing embryos. Therefore, we believe that we could distinguish between specific developmental and regenerative processes at 48 hpf examined in this study.

  1. Olig2 does not only label motor neuron progenitors, but also oligocytes and OPCs. There is a developmental switch from motorneurogenesis to oligodendrogenesis that starts a few hours before they lesion (see works of the Appel lab). 

Authors’ response to 5:

It is true that Olig2 labels motor neuron progenitors and both oligocytes and oligodendrocyte precursor cells, as you remarked. However, in order to further confirm our hypothesis that anp32a overexpression induces the increase of motor neuron, we injected BrdU into the brain and then performed SCI at 48 hpf for embryos derived from the cross strain between Tg(gfap:GFP), in which radial glial cells were tagged by GFP, and Tg(mnx1:TagRFP), in which motor neurons were tagged by RFP. As shown in Figure 8, the number of cells displaying a BrdU-positive signal (purple) and also colocalized with GFAP-positive (green, Figures 8C, D) and mnx-positive (red, Figures 8E, F) signals was increased in SCI-embryos overexpressing Anp32a at the SCI site, suggesting that Anp32a overexpression increased both the number of radial glial cells and motor neurons. Therefore, we conclude that overexpression of Anp32a induces to enhance the proliferation of motor neurons progenitors and radial glial cells, which, in turn, the increased of motor neurons. Consequentially, this outcome may contribute to neuronal regeneration in the spinal cord of SCI-embryos (see new description in the main text, lines 1 to 12 of page 16).

  1. Migration of mnx1:gfp positive neurons into a spinal injury has been described, how do the authors distinguish between axons and neurons if they just look at fluorescence levels in the injury site? There is imprecision in the language throughout between neuronal and axonal regeneration. Furthermore, the authors do not formally show that the neurons in the injury site are new. EdU experiments would have to be performed to state this with confidence, especially since it was recently reported that pre-existing neurons can migrate into the injury site in early phases of regeneration (Vandestadt et al Dev Cell 2021).

Authors’ response to 6:

Above, we described how we injected BrdU into the brain and then performed SCI at 48 hpf in embryos derived from the cross strain between Tg(gfap:GFP) and Tg(mnx1:TagRFP). We further described how cells that showed BrdU-positive signals, also colocalizing with GFAP-positive and mnx-positive signals, increased their numbers in SCI-embryos overexpressing Anp32a at the SCI site. Again, these results suggest that Anp32a overexpression increased both the number of newly formed radial glial cells and motor neurons.

                 Thank you for reminding us to correctly distinguish between neuronal and axonal regeneration. We revised accordingly.

  1. In the analyses presented in Figure 4, the wrong statistics have been applied, since the authors are using the same groups for multiple comparisons. An ANOVA with a posthoc test would be more appropriate. This is mentioned in the statistics, but not in the figure legends.

 Authors’ response to 7:

To correct our mistake in revised Figure 4C, we employed one-way ANOVA, followed by Tukey’s multiple comparisons test (please see table below). The statistical analysis of Figure 4 was corrected as well. (Please see lines 25 to 26 of page 8).

  1. Methods to reduce bias have been insufficiently discussed (apart from randomisation).

Authors’ response to 8:

In response to your comment, we added more detailed descriptions in the Material and methods section as follows: “To reduce biased data from the transgenic studies during each experiment, we first chose normally developed embryos to perform gene transfer. After transgenesis, we discarded transgenic embryos exhibiting bizarre appearance, and then randomized to select the examined samples, ruling out samples displaying extreme values, such as too high and too low. Transgenic embryos that exhibited abnormal expression patterns, compared to control, were also deleted. Each trial resulted in a random selection repeated at least three to five times to obtain one single average. Then, the final data presented in the manuscript was averaged from three independent experiments, followed by the appropriate statistical analysis. Last, apart from precise definition of the SCI-site, we also tried to increase the sample size in order to reduce individual differences. For example, in figure 5B and 5C, 11 SCI-embryos were analyzed in the 18-hpi group, and 11 SCI-embryos were analyzed in the anp32a mRNA plus SCI group, while 10 SCI-embryos were analyzed in the 24-hpi group, and 10 SCI-embryos were analyzed in the anp32a mRNA plus SCI group. (Please see lines 9 to 21 of page 19).

            For Western blotting, we only took into account the intensity of Anp32a when       the expressional intensity of internal control, alpha-tubulin, loaded into each well on gel was almost identical on the same blot paper. (Please see lines 1 to 3 of page 20).

           We added the above details in the revised Materials and methods.

  1. The specimens in Figure 8 C and F are the same fish - this should be pointed out clearly.

Authors’ response to 9:

Yes, the specimens in Figures 8 C and F were originated from the same fish. We added this description in Materials and methods, as well as legends, if needed.

minor:

  1. „lower vertebrate“ is an inaccurate designation for fishes and amphibians. “Many anamniotes” or “many fishes and amphibians” can be used instead. The “many” is needed because post-metamorphic and adult frogs cannot regenerate spinal cord. Same for “higher vertebrate” used further down the text. 

Authors’ response to 10:

In response, we have corrected, as follows: (1) “Nevertheless, unlike mammalian organisms, many fishes and amphibians, such as zebrafish, can regenerate after TSCI treatment.” (Please see line 15 of page 2) and (2) “Anp32a found in many vertebrates is involved in diverse physiological functions.” (Please see lines 50 to 51 of page 2).

  1. In the sentence “Furthermore, no permanent glial scar formation does not” there is a double negative

    Authors’ response to 11:

In response, we revised as follows: “Furthermore, glial scar formation of zebrafish is restricted after SCI, resulting in favoring rapid axonal regeneration and the improvement of neuronal regeneration of SCI- zebrafish.” Please see lines 20 to 22 of page 2.

  1. “the absence of immune rejection during transplantation makes Zebrafish the dominant model system for real-time screening of transplanted cells harboring genes vital for successful regeneration after SCI.“Ma et al did not transplant any cells and I am personally only aware of the present authors who have transplanted cells into the zebrafish spinal cord after injury. It is therefore puzzling that the authors would describe the zebrafish as the “dominant model system” for such approaches.

Authors’ response to 12:

In response to your comment, we revised this statement, as follows: “Recently, recovery function of many important cellular and molecular factors in-volved in axonal regeneration have been identified using zebrafish as an in vivo model. For example, Ma et al. (2014) [17] found that the legumain is essential for functional recovery after SCI in zebrafish. Reimer et al. (2008) [18] showed that new motor neurons originating from proliferating Olig2+ ependymo-radial glial cells mature and eventually form synaptic connections. Mokalled et al. (2016) [19] found that connective tissue growth factor is necessary and sufficient to stimulate glial bridging and natural spinal cord regeneration. Wehner et al. (2017) [20] identified that Wnt signaling controls pro-regenerative Collagen XII and thereby contributes to ECM that is growth-promoting for axons at the injury site. Moreover, the absence of immune rejection during trans-plantation makes zebrafish a promising model system for real-time screening of trans-planted cells harboring genes vital for successful regeneration after SCI [21]. Thus, zebrafish is increasingly popular for studying neuronal regeneration after SCI.” (Please see lines 23 to 35 of page2).

  1. “The spinal cord between 17-20 th somite developed at 48 hpf, which is located above the anus of embryos from Tg(gfap:GFP), was transected by tweezer.“ It is not clear what is meant here – can the author rephrase.

Authors’ response to 13:

In response, we rephrased, as follows: “The spinal cord between the 17th and 20th somite located above the anus of Tg(gfap:GFP) embryos at 48-hpf was performed by mechanical crush injury to cause SCI.”  

In the section subtitled “4.2. Mechanical crush injury to cause SCI of zebrafish larvae”, we presented the following: “We followed the protocol described by Zeng et al. (2021). Briefly, embryos at 48 hpf were immersed in embryo medium containing 0.02% tricaine for anesthesia. A tweezer (Dumont No.5) was employed to perform spinal cord crush lesion between the 17th and 20th somite located above the anus to cause SCI. These SCI-embryos were placed in fresh medium containing 0.003% 1-phenyl-2-thiourea for study.” (Please see lines 37 to 42 of page 18)

  1. “treated with SCI” – spinal cord injury is not a treatment. 

Authors’ response to 14:

Thank you. We made the appropriate correction throughout the revised paper.  

  1. Proliferation studies: it is not clear that neural cells proliferate, clearly this would not be post-mitotic neurons (as labelled by the line in figure 5).  A spinal cord injury leads to the influx of a significant number of non-neural cells into the injury site (immune cells, fibroblasts, keratinocytes) – which cells proliferate?

Authors’ response to 15:

It’s true that spinal cord injury leads to the influx of a significant number of non-neural cells to the injury site. Actually, in the results shown in Figure 5 only provided us a clue that neural cells might proliferate after SCI. To confirm this hypothesis, we went further to perform FACS study. The results shown in Figures 6 and 7 did demonstrate that Anp32a could improve the proliferation of motor neuron progenitor cells and radial glial cells at the SCI site of SCI-embryos. Furthermore, as shown in Figure 8, overexpression of Anp32a in SCI-embryos from crossing Tg(gfap:GFP) and Tg(mnx1:TagRFP) displayed the increase of both GFP+/BrdU+ and RFP+/BrdU+ signals at the lesion site compared to control embryos. This evidence suggested that, at the very least, motor neurons and radial glial cells were actively proliferated at the injury site and that such increase is induced by Anp32a overexpression. The proliferation of non-neural cells owing to Anp32a overexpression may be an avenue of investigation in the future. Please see lines 39 to 51 of page 15.

  1. Reference 45 does not use an antibody against olig2 – how was the antibody used in FACS experiments validated?

Authors’ response to 16:

We apologized for providing insufficient information. We revised the statements as follows: The FACS protocol was followed by Brody et al. [75] (not Ref. 45 as you mentioned above), while the protocol of using olig2 immunostaining in FACS was followed by Zeng et al. [76].

In response to your question, we gave more detail about our use of olig2-antiserum in the FACS experiment and revised accordingly in Methods, as follows: “Using antibody against olig2 in FACS, we followed the protocol described by Zeng et al. [76] with some modifications. The anp32a mRNA was microinjected into the one-cell stage of embryos from wild-type or Tg(gfap:GFP), followed by SCI at 48 hpf and deheading.  Cells were suspended and stained with antibody against olig2 and DAPI at 24 hpi. We randomly collected 100 SCI-embryos, suspended cells from deheaded embryos, fixed, and performed cell immunostaining by using Olig2 antibody labeled with RFP or DAPI staining. After analyzing the population of cells, the percentage of GFP/RFP-positive cells expressing DAPI signals was determined. Please see lines 26 to 33 of page 19.

Reviewer 2 Report

The present study by Lee et al. investigated the possible role of Acidic nuclear phosphoprotein 32 family, member A (ANP32A) in the neuronal regeneration of zebrafish following spinal cord injury (SCI). By  utilizing gain- and loss-of-function strategies, as well as fluorescence-activated cell sorting and BrdU-labeling in two transgenic zebrafish lines (Tg(mnx 1:GFP); Tg(gfap:GFP)), the authors could show that ANP32A overexpression enhances the regeneration of motor neurons and swimming capability in SCI-treated zebrafish larvae, whereas ANP32A knock-down showed the opposite pattern. Immunofluorescence staining in combination with fluorescence-activated cell sorting (FACS) showed that ANP32A could facilitate cell proliferation and promotes the proliferation of radial cells to aid in the regeneration of motor neurons following SCI. The results support a link between ANP32A and proliferation of neural cells in SCI-treated zebrafish embryos, possibly through the activation of signal transducer and activator of transcription (STAT) signaling in radial glial cells and motor neurons during regeneration. The study is of interest for the readers of the International Journal of Molecular Sciences, also specifically because the role of ANP32A is largely unknown in zebrafish. However, the manuscript should be revised with a focus on following issues:

(1) In Figure 1 the authors show the time course of SC regeneration after SCI. In C, they quantified the distance of the two sites and performed a Student’s t-test as statistical analysis. Could the authors elaborate on the total number of animals used (show individual data points in the graph) and why they decided to perform a t-test as statistical analysis? In my opinion, a repeated measure ANOVA would be more suitable. 

(2) As loss-of-function strategy, the authors designed an antisense morpholino (ANP32A-MO) to block the translation of endogenous anp32a mRNA. To rescue ANP32A (gain-of-function strategy), they designed a wobble-ANP32A-flag mRNA. Based on the image of the Western Blot (Figure 2), endogenous ANP32A seems almost non-present. Did the authors quantify the band intensities? What could be the possible explanation? 

(3) Although the authors state that they are employing zebrafish embryos from transgenic line Tg(mnx1:GFP) in which GFP is specifically labeled in motor neurons, I am wondering about which cells are targeted by the antisense morpholino (ANP32A-MO)? All cells or is there any cell-specificity?       

(4) Did the authors perform any dose-response assays of ANP32A-MO (or wobble-ANP32A-flag mRNA) to determine the margin between the induction of a specific phenotype and the onset of toxicity?

(5) To study the role of ANP32A on the spinal cord regeneration process after SCI, the authors injected ANP32A-MO or wobble-ANP32A-flag mRNA. While the gain-of-function data looks strong (Figure 3), I am uncertain whether the results of the loss-of-function experiment are convincing. Multiple sources (e.g., Corey & Abrams, 2001; ...) discuss that when blocking a gene of unknown function, it is essential to bear in mind that the new phenotype (e.g., Figure 4) is not necessarily due to reduction of expression of the target gene, that antisense technology rarely duplicates complete 'loss-of-function' mutations, and that multiple controls (e.g., mismatch and scrambled control oligomer) are essential in order to draw robust conclusions. Although the authors performed mRNA rescue experiments, I am wondering whether the results obtained in the experiments were ever confirmed by comparison to phenotypes of existing mutants.

(6) Figure 4: Did the authors check for a normal distribution of the data, as well as for potential outliers? If not normally distributed, a Student’s t-Test is not the right statistical analysis to check for differences between groups. Moreover, authors should consider performing a 1-Way ANOVA, rather than repeated t-Tests.    

(7) Figure 4: Along the lines of the previous comment, I find it interesting that there is a highly significant difference between the +,-,-,- group and the +,+,-,- group but apparently not between the +,+,-,- group and the +,-,+,- group. Does this still hold up when performing a 1-Way ANOVA?       

(8) To test whether ANP32A confers neural cells with increased proliferation during the neuronal regeneration, embryos were injected with ANP32A mRNA, treated with SCI and immunostaining for the proliferation marker phosphor-histone 3 (PH3) was performed (Figure 5). Based on the images Fig. 5B, the number of PH3-positive cells in both groups (for both time-points) looks more or less similar, only that the ROI is slightly shifted. Could it be that the expression pattern (but not the number) of the proliferation marker differed?      

(9) Figure 8: I again have an issue with the data analysis. For A + B a 1-way ANOVA should be used. Data points for D and F do not seem normally distributed.         

(10) In their current study, the authors investigate the role of ANP32A in neuronal regeneration in Zebrafish. Why did the authors focus on ANP32A? Would the authors expect to see similar effects of ANP32B-E?

(11) Minor Comments:

(a) The resolution of all figures is very poor and should be re-done.

(b) Material and Methods should be briefly described and not just referred to a previous paper.

(c) Potential typo in the Figure Caption of Figure 4, page 8: ‘… Datum of each group …’. I assume that the authors mean ‘data’.

Author Response

Reviewer 2

The present study by Lee et al. investigated the possible role of Acidic nuclear phosphoprotein 32 family, member A (ANP32A) in the neuronal regeneration of zebrafish following spinal cord injury (SCI). By  utilizing gain- and loss-of-function strategies, as well as fluorescence-activated cell sorting and BrdU-labeling in two transgenic zebrafish lines (Tg(mnx 1:GFP); Tg(gfap:GFP)), the authors could show that ANP32A overexpression enhances the regeneration of motor neurons and swimming capability in SCI-treated zebrafish larvae, whereas ANP32A knock-down showed the opposite pattern. Immunofluorescence staining in combination with fluorescence-activated cell sorting (FACS) showed that ANP32A could facilitate cell proliferation and promotes the proliferation of radial cells to aid in the regeneration of motor neurons following SCI. The results support a link between ANP32A and proliferation of neural cells in SCI-treated zebrafish embryos, possibly through the activation of signal transducer and activator of transcription (STAT) signaling in radial glial cells and motor neurons during regeneration. The study is of interest for the readers of the International Journal of Molecular Sciences, also specifically because the role of ANP32A is largely unknown in zebrafish. However, the manuscript should be revised with a focus on following issues:

  • In Figure 1 the authors show the time course of SC regeneration after SCI. In C, they quantified the distance of the two sites and performed a Student’s t-test as statistical analysis. Could the authors elaborate on the total number of animals used (show individual data points in the graph) and why they decided to perform a t-test as statistical analysis? In my opinion, a repeated measure ANOVA would be more suitable. 

Authors’ response to (1):

Thank you for pointing out the error. In the revised Figure 1C, we employed one-way ANOVA, followed by Tukey’s multiple comparisons test (please see table below).

Additionally, we also presented individual data points in the graph as shown in revised Figure 1C (please see figure below).

(2) As loss-of-function strategy, the authors designed an antisense morpholino (ANP32A-MO) to block the translation of endogenous anp32a mRNA. To rescue ANP32A (gain-of-function strategy), they designed a wobble-ANP32A-flag mRNA. Based on the image of the Western Blot (Figure 2), endogenous ANP32A seems almost non-present. Did the authors quantify the band intensities? What could be the possible explanation? 

Authors’ response to (2):

Our data demonstrated that the endogenous Anp32a level during 48 hpf was low. Therefore, we hypothesized that Anp32a -mediated cell proliferation might not be quite activated in untreated embryos at 48 hpf compared to the SCI-embryos, suggesting that Anp32a is important for motor neurons and radial glial cells regeneration after SCI. 

(3) Although the authors state that they are employing zebrafish embryos from transgenic line Tg(mnx1:GFP) in which GFP is specifically labeled in motor neurons, I am wondering about which cells are targeted by the antisense morpholino (ANP32A-MO)? All cells or is there any cell-specificity?       

Authors’ response to (3):

As far as we know, exogenous MO was distributed throughout the body of microinjected embryos at the one-cell stage. Therefore, we believe that no specific cells would be targeted by anp32a-MO.

(4) Did the authors perform any dose-response assays of ANP32A-MO (or wobble-ANP32A-flag mRNA) to determine the margin between the induction of a specific phenotype and the onset of toxicity?

 Authors’ response to (4):

This is a critical question for scientists who employ loss-of-function to determine gene function, i.e., whether the phenotypes of MO-injected embryos result from specific induction or toxicity of injected MO. As shown in Figures 2~4 in this study, we first performed a dose-responsive assay of anp32a-MO. We examined 1, 1.5, 1.75, 2 and 4 ng prior to each 2.3 nl injection for the knockdown experiment. Results demonstrated that the phenotypes induced by injected MO were dose-dependent. Among them, 2 ng was the most suitable concentration for further study. Second, we injected wobble-anp32a-flag mRNA to see whether the phenotypes of MO-injected embryos had been specifically rescued by co-injection of anp32a-flag mRNA in a dose-dependent manner. Here, results showed that the rescue effect of spinal cord regeneration was increased when the concentration of anp32a-flag mRNA was increased from 200 to 400 pg. Also, the minimal concentration of anp32a-flag mRNA that could specifically rescue MO-induced defects was 200 pg. This line of evidence suggests that the defective phenotypes induced by the concentration of anp32a-MO used to knock down anp32a in this study did not result from the onset of toxicity. (Please see line 46 of page 17 to line 7 of page 18 in Discussion section).

(5) To study the role of ANP32A on the spinal cord regeneration process after SCI, the authors injected ANP32A-MO or wobble-ANP32A-flag mRNA. While the gain-of-function data looks strong (Figure 3), I am uncertain whether the results of the loss-of-function experiment are convincing. Multiple sources (e.g., Corey & Abrams, 2001; ...) discuss that when blocking a gene of unknown function, it is essential to bear in mind that the new phenotype (e.g., Figure 4) is not necessarily due to reduction of expression of the target gene, that antisense technology rarely duplicates complete 'loss-of-function' mutations, and that multiple controls (e.g., mismatch and scrambled control oligomer) are essential in order to draw robust conclusions. Although the authors performed mRNA rescue experiments, I am wondering whether the results obtained in the experiments were ever confirmed by comparison to phenotypes of existing mutants.

Authors’ response to (5):

We appreciate your comments regarding the use of MO to perform loss-of-function. Although the use of mutants would be more convincing, no zebrafish Anp32a mutant is currently available. Despite some concerns over using the MO-base gene knockdown approach, Morpholino oligomers remain an essential tool to transiently inhibit gene expression in the zebrafish model, as attested by the numerous MO-related papers published by our lab (Lin et al., 2013, 2022; Lee et al., 2015; Fu et al., 2017). As noted above, we employed MO-knockdown combined with rescue through anp32a mRNA and Western blot analysis to demonstrate that the defective phenotype of spinal cord regeneration induced by anp32a -MO was specific. (Please see lines 8 to 14 of page 18 in Discussion section)

(6) Figure 4: Did the authors check for a normal distribution of the data, as well as for potential outliers? If not normally distributed, a Student’s t-Test is not the right statistical analysis to check for differences between groups. Moreover, authors should consider performing a 1-Way ANOVA, rather than repeated t-Tests.    
Authors’ response to (6):

Thank you for pointing out the error. In the revised Figure 4, we employed one-way ANOVA, followed by Tukey’s multiple comparisons test (please see table below).

(7) Figure 4: Along the lines of the previous comment, I find it interesting that there is a highly significant difference between the +,-,-,- group and the +,+,-,- group but apparently not between the +,+,-,- group and the +,-,+,- group. Does this still hold up when performing a 1-Way ANOVA?       

Authors’ response to (7):

Thank you for your suggestion. Indeed, after we corrected by using one-way ANOVA, followed by Tukey’s multiple comparisons test between the +,+,-,- (RNA+SCI) group and the +,-,+,- (MO+SCI) group, we found a highly significant difference between these two groups  (p<0.001).(Please see table below)

(8) To test whether ANP32A confers neural cells with increased proliferation during the neuronal regeneration, embryos were injected with ANP32A mRNA, treated with SCI and immunostaining for the proliferation marker phosphor-histone 3 (PH3) was performed (Figure 5). Based on the images Fig. 5B, the number of PH3-positive cells in both groups (for both time-points) looks more or less similar, only that the ROI is slightly shifted. Could it be that the expression pattern (but not the number) of the proliferation marker differed?      

Authors’ response to (8):

     In response, we demonstrated that the expression patterns of the proliferation marker PH3 between these two groups at the lesion site were different, as figures shown below.

Additionally, statistical analysis of the number of PH3-positive cells presented at the SCI site of anp32a mRNA-injected embryos at 18 hpi was significantly increased, as shown in Figure 5C. (Please see figure below)

(9) Figure 8: I again have an issue with the data analysis. For A + B a 1-way ANOVA should be used. Data points for D and F do not seem normally distributed.         
Authors’ response to (9):

We used the corrected statistical analysis, as suggested, to perform data analysis. In the revised Figure 8B, we used one-way ANOVA followed by Tukey’s multiple comparisons test to perform statistical analysis. Please see lines 11 to 12 of page 14.

(10) In their current study, the authors investigate the role of ANP32A in neuronal regeneration in Zebrafish. Why did the authors focus on ANP32A? Would the authors expect to see similar effects of ANP32B-E?

Authors’ response to (10):

In response to your suggestion, we added our rationale for studying anp32 in the revised Introduction. In summary, we previously used transcriptomic analysis to screen key regulatory genes involved in neuronal regeneration after spinal cord injury (SCI) of zebrafish embryos (Zeng et al., 2021). Among 3911 candidate genes, they categorized all of them into four groups based on Gene Ontology. In our study, we focused on candidate genes in the regeneration/development-related genes group and selected runx3, elav11A, crabp1a, bace1, inpp5e and anp32a. Among these candidates analyzed in preliminary studies, only knockdown of either inpp5e or anp32a could significantly impair swimming ability in SCI-embryos (see attached figures below). Furthermore, the impairment induced by anp32a-MO was more serious than that induced by inpp5e-MO. Therefore, we chose anp32a for further study. We added this detailed information in the Introduction (Please see lines 36 to 45 of page 2).

Whether other ANP32 family proteins, such as ANP32B and ANP32E, also play important roles in neuronal regeneration in zebrafish SCI-embryos, although they are not focused in the present study, is worthy to study in the further.

(11) Minor Comments:

(a) The resolution of all figures is very poor and should be re-done.

 Authors’ response to (a):

Thank you for your comments. The quality of all figures was improved in the revised manuscript.

(b) Material and Methods should be briefly described and not just referred to a previous paper.

Authors’ response to (b):

In response to your suggestion, we added more description in the Materials and Methods section of the revised manuscript. (Please see Materials and Methods section)

(c) Potential typo in the Figure Caption of Figure 4, page 8: ‘… Datum of each group …’. I assume that the authors mean ‘data’.

Authors’ response to (c):

Thank you. We correct this mistake in the revised manuscript. (Please see line 23 of page 8).

Round 2

Reviewer 2 Report

The authors have satisfactorily responded to all my questions and made the necessary changes to the manuscript.

Author Response

English language and style are fine checks required.

Author's response:

We do our best to edit for English content with the help of a native English-spoken specialist. The revised words and sentences were marked in red in the revised manuscript.
